# Physical Stimulation Methods Developed for In Vitro Neuronal Differentiation Studies of PC12 Cells: A Comprehensive Review

**DOI:** 10.3390/ijms25020772

**Published:** 2024-01-07

**Authors:** Kanako Tominami, Tada-aki Kudo, Takuya Noguchi, Yohei Hayashi, You-Ran Luo, Takakuni Tanaka, Ayumu Matsushita, Satoshi Izumi, Hajime Sato, Keiko Gengyo-Ando, Atsushi Matsuzawa, Guang Hong, Junichi Nakai

**Affiliations:** 1Division of Oral Physiology, Tohoku University Graduate School of Dentistry, Sendai 980-8575, Japan; 2Laboratory of Health Chemistry, Graduate School of Pharmaceutical Sciences, Tohoku University, Sendai 980-8578, Japan; 3Cell Resource Center for Biomedical Research, Institute of Development, Aging and Cancer, Tohoku University, Sendai 980-8575, Japan; 4Graduate School of Life Sciences, Tohoku University, Sendai 980-8577, Japan; 5Division for Globalization Initiative, Tohoku University Graduate School of Dentistry, Sendai 980-8575, Japan; 6Division of Pharmacology, Meikai University School of Dentistry, Sakado 350-0283, Japan

**Keywords:** PC12, neuronal differentiation, physical stimulation

## Abstract

PC12 cells, which are derived from rat adrenal pheochromocytoma cells, are widely used for the study of neuronal differentiation. NGF induces neuronal differentiation in PC12 cells by activating intracellular pathways via the TrkA receptor, which results in elongated neurites and neuron-like characteristics. Moreover, the differentiation requires both the ERK1/2 and p38 MAPK pathways. In addition to NGF, BMPs can also induce neuronal differentiation in PC12 cells. BMPs are part of the TGF-β cytokine superfamily and activate signaling pathways such as p38 MAPK and Smad. However, the brief lifespan of NGF and BMPs may limit their effectiveness in living organisms. Although PC12 cells are used to study the effects of various physical stimuli on neuronal differentiation, the development of new methods and an understanding of the molecular mechanisms are ongoing. In this comprehensive review, we discuss the induction of neuronal differentiation in PC12 cells without relying on NGF, which is already established for electrical, electromagnetic, and thermal stimulation but poses a challenge for mechanical, ultrasound, and light stimulation. Furthermore, the mechanisms underlying neuronal differentiation induced by physical stimuli remain largely unknown. Elucidating these mechanisms holds promise for developing new methods for neural regeneration and advancing neuroregenerative medical technologies using neural stem cells.

## 1. Introduction

The rat PC12 cell line is one of the most common neuronal precursor cell lines derived from adrenal pheochromocytoma cells, which are the malignant counterpart to chromaffin cells. It has an embryonic origin from the neural crest and consists of a mixture of neuroblastic and eosinophilic cells. The PC12 cell line has been used as an excellent model for the neuronal differentiation of mammalian cells [1,2]. It can form synapses with primary cultured neurons isolated from the rat cerebral cortex [3]. PC12 cells differentiate into neuron-like cells when exposed to neurotrophic factors, including nerve growth factor (NGF) [2,4]. NGF binds to its membrane receptor, tropomyosin-related kinase A (TrkA), which is expressed in these cells. Thus, NGF can activate its target intracellular pathways in PC12 cells to induce neuronal differentiation [5,6] (Figure 1).

Treatment of PC12 cells with NGF triggers the activation of extracellular signal-regulated kinases 1 and 2 (ERK1/2), which are components of the mitogen-activated protein kinase (MAPK) family, through the activation of TrkA. The activation of ERK1/2 results in the elongation of neurites and the development of neuron-like characteristics in PC12 cells. NGF-induced differentiation also requires p38 MAPK, another member of the MAPK family, which may cooperate with ERK1/2 for the promotion of differentiation [4,7] (Figure 1).

Other than NGF, many molecules, including endogenous soluble factors and chemical substrates, can induce neuronal differentiation in PC12 cells [6]. For example, bone morphogenetic proteins (BMPs), such as BMP2 and BMP4, belong to the large transforming growth factor-β (TGF-β) cytokine superfamily, which regulates various biological processes, including neuronal development [8]. BMPs form a complex with type I and type II transmembrane receptors and activate two additional pathways: the TGF-β-associated kinase 1 (TAK1)–p38 MAPK signaling pathway and the Smad signaling pathway [9]. BMPs also stimulate neurite elongation in both PC12 cells and neurons [7]. BMP-induced neuritogenesis is mediated by p38 MAPK signaling in PC12 cells [10]. Nonetheless, the swift spread and brief lifespan of soluble factors such as NGF and BMPs may restrict their effectiveness within living organisms, including humans [7].

During the process of establishing PC12 cells as a model for in vitro neuronal cell differentiation, various subclones of PC12 cells were isolated that exhibited somewhat different characteristics compared with the parental cell line. These subclones were used for specific purposes related to neuronal differentiation in many cases. For example, using a single-cell cloning assay, Kudo et al. established the PC12-P1F1 cell line as a subclone of the PC12 cell line. This subclone exhibited hypersensitivity to thermal stimulation, which was used to study the effect of thermal stimulation on neuronal differentiation [7].

PC12 cells have also been used as a suitable in vitro model to examine the effects of various types of physical stimuli on neuronal cell differentiation since the cell line was established in 1976 [1,4]. However, the development of effective methods for inducing neuronal differentiation in PC12 cells through physical means and the understanding of the underlying molecular mechanisms is ongoing. A comprehensive review summarizing the progress of these studies has not been published. In this review, to support further advancements in this field, we procured and systematically classified the original research articles that examined the effects of various physical stimuli on neuronal differentiation in PC12 cells. The following sections begin with a brief overview of the various physical stimulation methods that have been examined, such as mechanical, ultrasound, electrical, magnetic, thermal, and optical stimuli. Subsequently, we discuss how these physical methods influence the neurite outgrowth process and neuronal differentiation in PC12 cells.

## 2. Effects of Various Physical Stimuli on the Neuronal Differentiation of PC12 Cells

### 2.1. Mechanical Stimulation

The mechanical environment plays an important role in various tissues of the human body. It is well established that various types of mechanical stress, such as shear stress, hypergravity, vibration, and gravity, act as regulators of cellular processes [11,12,13]. For example, shear stress affects a wide range of cellular phenomena, including growth and differentiation [14,15]. However, there have been only a few studies involving mechanical stimulation that evaluated the effects of shear stress, strain, or vibration on the differentiation of PC12 cells (Table 1). Among these, Kim et al. determined the effects of shear stress on neurite outgrowth in cells and found that shear stress of 0.25 Pa promoted NGF-induced neurite length [16]. In addition, Haq et al. showed that PC12 cells exposed to specific strains (4% at 1.0 Hz or 16% at 0.1 Hz) significantly increased the neurite length induced by NGF [17].

Vibration stimulation is one mechanical stimulus that has therapeutic effects in humans, particularly for muscle dysfunction [18,19,20]. Ito et al. studied the effects of nano-vibration on the differentiation of PC12 cells [21]. They showed that vibration at 10 kHz accelerated the NGF-induced neurite outgrowth in PC12 cells cultured in a serum-free medium.

Gravity is another common mechanical stimulus to which living cells are constantly subjected on Earth. Altered gravity, such as microgravity in space or hypergravity that can affect astronauts during the initial stages of travel from Earth to space, may have long-term effects on human health. With respect to PC12 cells, Genchi et al. studied the effects of hypergravity on neuronal cell differentiation [22]. They found that treatment with 150 g hypergravity significantly increased neurite outgrowth in cells exposed to NGF.

As described above, all types of mechanical stimulation discussed in this review suggest that physical stimuli can enhance the effects of NGF during neuronal cell differentiation in PC12 cells. However, the results of these studies also show that mechanical stimulation alone fails to induce neuronal differentiation in PC12 cells. Thus, it will be valuable in future studies to clarify whether a specific mechanical stimulation alone can induce neuronal differentiation in PC12 cells.

**Table 1 ijms-25-00772-t001:** The enhancing effect of mechanical stimulation on NGF-induced neuronal differentiation in PC12 cells.

Mechanical Stimulation	Mechanical Stimulation Conditions	Stimulation Term	Additional Condition for Inducing Differentiation	Detected Changes during Differentiation	Reference
Shear stresses	0.25 Pa	Total 6 h/day	NGF	a	[16]
Mechanical strain	Strain level: 4%, strain rate: 1.0 Hz	96 h	NGF	a	[17]
	Strain level: 16%, strain rate: 0.1 Hz	96 h	NGF		
Nano-vibration	10 kHz	1 h/day	NGF	a	[21]
Hypergravity	150 g	1 h	NGF	a, b	[22]

NGF: never growth factor. a: morphological changes with neuritogenesis. b: changes in gene expression.

### 2.2. Ultrasound Stimulation

Ultrasound can stimulate tissues by transmitting mechanical energy in a noncontact manner. Because it is generally challenging to apply mechanical stimulation noninvasively inside living tissues, the use of ultrasound for stimulation has gained significant attention in preclinical and clinical research.

Ultrasound is defined as sound waves with frequencies above the human hearing threshold [23]. Generally, an ultrasound wave has a frequency ranging from 1 to 12 MHz. When ultrasonic waves are applied to the human body, both thermal and nonthermal effects can occur. In contrast to thermal effects, in which energy is converted into heat because of the attenuation of ultrasound waves propagating through a living organism, nonthermal effects include cavitation, which generates mechanical vibrations, and a wide range of other effects, such as radiation pressure and acoustic flow, associated with the propagation of sound waves. Which of these effects is predominant depends on the intensity and frequency of the ultrasound energy (Table 2).

As an example, low-intensity pulsed ultrasound stimulation (LIPUS), which is both nonthermogenic and noninvasive to tissues, is delivered at low intensity (<0.1 W/cm^2^) and constant frequency (1–1.5 MHz). LIPUS exerts some physiological effects on tissues, such as promoting bone regeneration [24]. In another study, noncontact stimulation by ultrasound caused the oscillation of membrane proteins or the cell membrane, which could be a biological signaling factor [25,26]. The NGF receptor in PC12 cells is a transmembrane protein [6]; thus, it can be stimulated by oscillation of the cell membrane through ultrasound [25,26].

Some researchers have shown that LIPUS promotes NGF-induced neurite outgrowth. For example, Zhao et al. found that it could significantly enhance the neurite length of PC12 cells in the presence of NGF [27,28]. These studies also found that LIPUS activates the NGF-induced ERK1/2–CREB–Trx-1 pathway through stretch-activated ion channels to promote neurite outgrowth. In addition, Maruyama et al. demonstrated that ultrasound stimulation under certain conditions increased NGF-induced neurite lengths and controlled the direction of neurite outgrowth [29]. Hoop et al. reported the interesting observation that ultrasound stimulation leads to neurite outgrowth, but with piezoelectric stimulation using the piezoelectric polymer polyvinylidene fluoride (PVDF) instead of NGF [30]. Generally, piezoelectric PVDF can generate electrical charges on its surface through acoustic stimulation, and it is already used as a substrate for wireless neural differentiation. After PC12 cells were grown on PVDF membranes, wireless stimulation by ultrasound enhanced neurite outgrowth compared with the results obtained through NGF treatment. Moreover, the authors suggested that stimulating PVDF using ultrasound may lead to NGF-independent calcium channel activation, resulting in neurite generation; however, the precise mechanism remains unclear. Based on the above studies, ultrasound stimulation may represent a method for the noninvasive induction of neuronal differentiation in PC12 cells.

**Table 2 ijms-25-00772-t002:** The enhancing effect of ultrasound stimulation on neuronal differentiation in PC12 cells in the presence of an additional condition.

Ultrasound Stimulation Conditions				Additional Condition for Inducing Differentiation	Detected Changes during Differentiation	Reference
Frequency	Intensity	PRF	Stimulation Term			
1 MHz	50 µW/m^2^	100 Hz (20% duty cycle)	10 min/2 days	NGF	a, c	[27]
1 MHz	30 or 50 µW/m^2^	100 Hz (20% duty cycle)	10 min/2 days	NGF	a, c	[28]
78 kHz	30 V	n. a.	72 h	NGF	a, b	[29]
132 kHz	80 W	n. a.	Total 50 min/day	Piezoelectric stimulation	a	[30]

PRF: pulse repetition frequency, NGF: never growth factor, n. a.: not applicable. a: morphological changes with neuritogenesis, b: changes in gene expression, c: changes in protein expression and/or activity.

### 2.3. Electrical Stimulation

Meng et al. reported that electrical stimulation is a useful method for promoting neurite outgrowth with a minimally invasive or noninvasive approach using various in vitro neuronal differentiation models such as PC12 cells [31]. Electrical activity is an inherent property of human tissues and is an important phenomenon in tissue regeneration and wound healing. Electrophysiology is a highly developed field with a long history, beginning with the work of Borgens et al. on the regeneration of frog limbs exposed to an electric current [32]. In PC12 cells, many reports have shown that proper electrical stimulation can significantly promote neurite growth and neuronal differentiation of these cells (Table 3).

For example, in 1991, Nakae et al. showed that in the absence of NGF, the “theta” (4–7 Hz electroencephalogram (EEG) rhythm) pattern electrical stimulation, which was known to elicit stable long-term potentiation in the CA1 region of the hippocampus and was designed with bursts of 4 pulses/40 ms at 5 Hz in a repetitive cycle, induced neurite outgrowth in PC12 cells [33]. Electrical stimulation was applied to the cells through 1 mm diameter platinum-rod electrodes placed at 15 mm intervals in the culture medium. The results indicated that electrical stimulation induced the differentiation of PC12 cells without the addition of NGF.

Manivannan et al. directly stimulated the neurite terminals of NGF-treated PC12 cells with electrical pulses from a glass micropipette, which was connected to an Ag–AgCl wire and placed extracellularly [34]. A single stimulus was delivered with a current pulse of 1–2 μA in amplitude and 1 ms in duration. In the case of repetitive stimulation, the pulse interval was 300 ms. Therefore, current pulses totaling 4 ms were applied. They also found that the electrical pulse stimulation of the terminals induced exocytosis accompanied by a prolonged length of neurite outgrowth after a few days, which was strongly dependent on the presence of extracellular Ca^2+^.

Recently, Kawamura et al. showed that electrically stimulated mutant PC12m3 cells, which do not exhibit neurite outgrowth in response to NGF, can extend neurites in the presence of NGF through activation of the p38 MAPK pathway with current square-wave pulses (10 Hz) of 100 mA through silver electrodes [35]. Furthermore, Bhattarai et al. showed that rectangular pulse wave potentials of 100 mV/cm, with a frequency of 20 Hz alone to PC12 cells using platinum electrodes, induced neurite outgrowth [36]. They also found that the neuronal cell differentiation was promoted upon electrical stimulation of the PPy nanorod (PPy-NR)-treated cells. The PPy-NR is a nanomaterial synthesized with uniform dimensions, implying that by passing through the cell membrane into the cytoplasm, the nanoparticles may have uniformly activated specific cell signaling pathways to influence cell differentiation.

In this context, generally, the approach of directly immersing electrodes into a cell culture medium on a culture plate as described above is the most common method for applying electrical stimulation to cells or tissues. It is a simple system that requires only two electrodes and a power source for a regular culture-plate-based system; however, it requires the use of noble metals or graphite as electrodes, such as platinum, gold, titanium, silver, or other alloys, to avoid corrosion. Researchers should also be aware of cytotoxicity resulting from redox products, heterogeneity of the extent of electrical stimulation of cells in different locations on the well of a regular culture plate, and the potential for electrophoresis.

Another approach has been described for cell stimulation that may be used with a three-electrode system, in which cells attach directly to the working electrode [31]. The three-electrode system consists of a working electrode, a counter electrode, and a reference electrode [37]. The system is the most common electrochemical system for accurately measuring potential and current. It uses a reference electrode that acts as a reference for measuring or controlling the working electrode potential without passing any current. For this approach, cells are first seeded and cultured on the surface of an electrode, which is then used as the working electrode. Because the surface of the working electrode is equipotential, all of the attached cells receive a uniform electrical stimulation. However, when a metal working electrode is used, one must be more careful to avoid unnecessary interference resulting from potential redox reactions. For example, Park et al. found that when PC12 cells were electrically stimulated with 250 mV constant potential or nonsinusoidal alternating current (AC) potential (biphasic rectangular pulse wave, 0.005 Hz) in the absence of NGF, neurites extended well on a gold nanoparticle (NP)-coated scaffold [38]. They also reported that the neuron-specific cytoskeletal proteins ΝF-200 and β-tubulin were increased depending on the electrical stimulation.

Indium tin oxide (ITO), which is a translucent conducting metal oxide, is also commonly used in electrodes for direct cell attachment [39]. ITO is known for its high electron affinity, low resistivity, high transparency in the visible light region, and high chemical stability [40]. Because of its conductive and transparent properties, ITO offers the possibility to visually investigate cell behavior upon electrical stimulation; however, ITO is usually used on 2D substrates and is not suitable for porous 3D scaffolds [31]. As an example, Kimura et al. discovered that the application of rectangular pulse wave potentials (100 mV peak-to-peak) with a frequency of 100 Hz to PC12 cells cultured on an ITO electrode induced neurite outgrowth in the absence of NGF, followed by the induction of c-fos expression, which is essential for cell differentiation [41]. They also found that the induction of neuronal differentiation through electrical stimulation alone requires calcium ion influx through L-type calcium ion channels with stretch-activated channel (SA channel) properties, which may be significantly involved in the protein kinase C (PKC) cascade. Similarly, they also showed that when rectangular pulse wave potentials were applied to the electrodes at amplitudes of 200 mV and 400 mV with frequencies of 50 Hz, 500 Hz, and 1 kHz, the PC12 cells were differentiated most significantly at 200 mV and 100 Hz [42]. Moreover, Chang et al. reported that sinusoidal AC electrical stimulation (100 Hz, 100 mV/mm) promoted NGF-induced neurite outgrowth in PC12 cells seeded on the bottom of collagen- and ITO-coated chip dishes. They further showed that PKC activation by electrical stimulation enhanced NGF-induced signaling, particularly through the MEK–ERK1/2 pathway [43].

Conductive polymers, which is a general term for organic polymers that conduct electricity or their composites with other materials, can serve as electrodes and substrates for electrical stimulation in cultured cells [44]. When a semiconductor, including the above conductive polymer, is used as an electric substrate for growing cells, the electric substrate is wired into a complete electric circuit and does not produce electrophoresis or any electrode reactions [31]. Cells on the electric substrate may be exposed to potential gradients on the substrate surface, and electromagnetic fields can have an effect if an alternating current is used. Conductive polymers have the potential to be endowed with properties such as biocompatibility, biodegradability, and porosity through the appropriate structural design and manufacturing processes [45].

For example, Schmidt et al. reported that PC12 cells cultured on oxidized polypyrrole (PPy) film with a resistance of approximately 1 kΩ, and electrically stimulated with a steady potential of 100 mV through the film, significantly enhanced NGF-induced neurite outgrowth [46]. PPy is a commonly used conductive polymer in the physical and biomedical fields. It exhibits excellent biocompatibility and mechanical compatibility with living tissues.

As another example using PPy or its composites, Liu et al. exposed NGF-treated PC12 cells to 10 Hz (2% duty cycle), 100 Hz (2% duty cycle), and 250 Hz (5% duty cycle) charge-balanced biphasic pulsed-current stimulation with an amplitude of 1 mA [47] using novel PPy composites. The novel composites included a large polyelectrolyte dopant known as poly(2-methoxy-5 aniline sulfonic acid) (PMAS). The pulse duty cycle was defined as the percentage of time during which the pulse was switched on in one cycle. As a result, the electrical stimulation with frequencies of 100 and 250 Hz delivered directly through the PPy/PMAS composite films significantly enhanced neuronal cell differentiation in the presence of NGF, whereas electrical stimulation with frequencies of 250 Hz initiated reversible neurite sprouting in the absence of NGF. Moreover, Weng et al. demonstrated that charge-balanced biphasic pulsed-current stimulation (250 Hz, 5% duty cycle) with an amplitude of 1 mA promoted neurite outgrowth and orientation of PC12 cells cultured on inkjet-printed PPy/Col scaffolds [48].

Jing et al. revealed that when an electric field of 30–80 mV/mm was applied to PC12 cells in the presence of NGF cultured on conductive fibers known as “POP”, which were fabricated by coating PPy onto parallel-aligned poly(L-lactide) fibers (hereafter, PLLA fibers), the electrical stimulation enhanced the differentiation efficiency and neurite outgrowth of the cells [49]. In addition to PPy, poly(3,4-ethylenedioxythiophene) (PEDOT) is also a widely used conductive polymer. PEDOT exhibits high transparency and good chemical stability. It is primarily used in the field of neural implants, which are implantable electronic devices that include microelectrode arrays designed to electrically connect the devices to the nervous system. The neural implants can stimulate neural activity in various neural tissues [37].

As an example, Hsiao et al. reported that when an electric field of 120 mV/mm was applied to PC12 cells cultured on PEDOT that was electrochemically coated on ITO electrodes in the presence of NGF, the elongation of neurite outgrowth was enhanced. They also observed neurites of the PC12 cells aligned along the direction of the photo-etched patterned electrodes [50]. In addition, Zhu et al. seeded PC12 cells on biomimetic PEDOT-coated ITO electrodes and applied charge-balanced biphasic pulsed electrical stimulation (250 Hz; 5% duty cycle; and 20, 40, or 60 mV) to the cells [51]. They showed that the electrical stimulation with an amplitude of 60 mV elongated neurites in the presence of NGF on the biomimetic PEDOT.

As an example of other conductive polymers, Borah et al. suggested that electrical stimulation with rectangular pulse wave potentials (10 Hz) of 500 mV/cm could significantly enhance neurite outgrowth in PC12 cells on conductive poly[2-methoxy-5-(2-ethyl-hexyloxy)-1,4-phenylene vinylene] (MEH-PPV)-based, Col-coated polycaprolactone (PCL) electrospun nanofiber meshes [52]. Moreover, Xu et al. developed silver nanowires (AgNWs) embedded with polydimethylsiloxane (PDMS), which exhibited good biocompatibility and generated a stable electric field during mechanical stretching [53]. Electrical stimulation with different amplitudes (60 mV, 120 mV, 240 mV) at a frequency of 20 Hz in combination with mechanical stretching stimulation, a tensile strain of 10%, and a frequency of 0.25 Hz promoted neuronal differentiation in PC12 cells on AgNW/PDMS electrodes.

Carbon is still widely used as an electrode material for electrochemistry. It is also known for its high corrosion resistance and inertness under various conditions such as strong acids and bases. Recently, nanocarbon materials such as carbon nanotubes (CNTs) and graphene have been used in semiconductor devices because of their high potential and wide range of applications. As an example of CNTs being used as a substrate for cell culture, Cho et al. showed that PC12 cells cultured on an electrically conductive composite made of collagen (Col) and CNTs in the presence of NGF enhanced the length of neurites after applying a constant voltage of 0.1 V for 6 h [54].

Furthermore, Sun et al. developed a biodegradable poly(caprolactone fumarate) (PCLF) scaffold through the incorporation of graphene, CNTs, and [2-(methacryloyloxy)ethyl]trimethylammonium chloride (MTAC), to create a conductive, positively charged scaffold [55]. They showed that electrical stimulation at 100 mV/mm with 20 Hz significantly promoted neurite outgrowth in PC12 cells cultured on the PCLF–graphene–CNT–MTAC scaffold in the presence of NGF.

In addition, Yang et al. developed functional annealed graphene oxide–collagen (aGO–Col) that included electroactive 3D crumpled structures [56]. They showed that aGO–Col composites enhanced the differentiation efficiency of PC12 cells in the presence of NGF. They also applied an electrical stimulation (biphasic square wave, 23.6 Hz) with 100 mV to PC12 cells cultured on the aGO–Col-coated retinal chip, which is a photovoltaic self-powered implantable chip. In the study, the electrical stimulation signal was generated by the retinal chip and controlled by the LED switch to cycle the lights, which were turned on for 1 min and off for 4 min for 1 h each day. They found that the electrical stimulation more prominently enhanced neuronal differentiation and neurite outgrowth on the aGO–Col-coated retinal chip.

In the electrical stimulation experiments that we introduced, the type of electrical signal transmitted was always the most important and complex element. These signals vary by direct current (DC) or alternating current (AC), field strength and direction, frequency and waveform, and duty cycle and may affect cells and tissues differently. On the other hand, the materials for electric substrates that serve as scaffolds for cell culture are extremely important, but material engineering in the medical field is still progressing with respect to biocompatibility, conductivity, and durability. Therefore, the development of new conductive polymers and composite materials such as CNTs has the potential to further advance this field.

**Table 3 ijms-25-00772-t003:** The enhancing effect of electrical stimulation on neuronal differentiation of PC12 cells and their derivatives in the presence or absence of an additional condition.

Electrical Stimulation Conditions				Scaffold, Electrode	Cell Line	Additional Condition for Inducing Differentiation	Detected Changes during Differentiation	Reference
Stimulation Pattern	Frequency	Intensity	Stimulation Term					
“Theta” (4–7 Hz EEG rhythm) pattern	Bursts of 4 pulses/40 ms, 5 Hz PRF	Maximum voltage: 2.5 V/cm	8–11 days	Platinum-rod electrodes	PC12	Not used	a	[33]
Current pulse	Pulse duration: 1 ms, pulse interval: 300 ms	1–2 μA	Total 4 ms	Glass/Ag-AgCl wire	PC12	NGF	a	[34]
Rectangular pulse	10 Hz	100 mA	30 min	Silver electrodes	PC12m3	NGF	a, c	[35]
Rectangular pulse	20 Hz	100 mV/cm	1 h/day	Platinum electrodes	PC12	Not used	a	[36]
						PPy-NRs		
Constant voltage	DC	250 mV	1 h/3 days	Gold nanoparticles coated scaffold	PC12	Not used	a, b	[38]
Biphasic rectangular pulse	0.005 Hz							
Rectangular pulse	100 Hz	100 mV	30 min/day	Poly-L-lysine-coated ITO electrode	PC12	Not used	a, b, c	[41]
Rectangular pulse	100 Hz	200 mV	30 min/day	Collagen-coated ITO electrode	PC12	Not used	a	[42]
Sinusoidal AC voltage	100 Hz	100 mV/mm	2 h,	Collagen and ITO-coated scaffold	PC12	NGF	a, c	[43]
Constant voltage	DC	100 mV	2 h	PPy films	PC12	NGF	a	[46]
Charge-balanced biphasic pulse	100 Hz (2% duty cycle), 250 Hz (5% duty cycle)	±1 mA	8 h/day	PPy/PMAS films	PC12	NGF	a	[47]
Charge-balanced biphasic pulse	250 Hz (5% duty cycle)	±1 mA	2 h	Inkjet-printed PPy/collagen scaffold:	PC12	NGF	a	[48]
Constant voltage	DC	30–80 mV/mm	2 h/day	PPy-coated PLLA fibrous mesh	PC12	NGF	a, b, c, d	[49]
Constant voltage	DC	120 mV/mm	48 h	PEDOT-coated ITO electrodes	PC12	NGF	a	[50]
Charge-balanced biphasic pulse	250 Hz (5% duty cycle)	60 mV	8 h/day	Biomimetic PEDOT-coated ITO electrodes	PC12	NGF	a	[51]
Rectangular pulse	10 Hz	500 mV/cm	2 h/day	MEH-PPV-based collagen coated PCL electrospun nanofiber meshes	PC12	NGF	a	[52]
Rectangular pulse	20 Hz	60, 120, 240 mV	4 h/day	AgNW/PDMS electrodes	PC12	not used	a	[53]
		240 mV				mechanical stretching stimulation		
Constant voltage	DC	0.1 V	6 h	CNT/collagen composites	PC12	NGF	a	[54]
Sinusoidal AC voltage	20 Hz	100 mV/mm	2 h/day	PCLF–Graphene–CNT–MTAC scaffolds	PC12	NGF	a	[55]
Biphasic rectangular pulse	23.6 Hz, (on term: 1 min, off term: 4 min)	100 mV	1 h/day	aGO–collagen-coated retinal chip	PC12	NGF	a	[56]

EEG: electroencephalogram; PRF: pulse repetition frequency; NGF: never growth factor; DC: direct current; ITO: indium tin oxide; AC: alternate current; PPy: polypyrrole; PMAS: poly(2-methoxy-5 aniline sulfonic acid); PLLA: poly(D,L-lactide); PEDOT: poly(3,4-ethylenedioxythiophene); MEH-PPV: poly [2-methoxy-5-(2-ethyl-hexyloxy)-1,4-phenylene vinylene]; PCL: polycaprolactone, AgNW: silver nanowire; PDMS: polydimethylsiloxane; CNT: carbon nanotube; PCLF: poly(caprolactone fumarate); MTAC: [2-(methacryloyloxy)ethyl]trimethylammonium chloride; aGO: annealed graphene oxide. a: morphological changes with neuritogenesis. b: changes in gene expression. c: changes in protein expression and/or activity. d: changes in other factors.

### 2.4. Electromagnetic Field Stimulation

Electromagnetic waves is a general term for energy waves generated by electric and magnetic fields. They are classified into ionizing radiation and nonionizing radiation depending on their frequency and energy [57]. Ionizing radiation consists of subatomic particles that have sufficient energy to ionize atoms or molecules by detaching electrons from them. The frequency of ionizing radiation is electromagnetic radiation above 3 PHz, which includes cosmic rays, gamma rays, and X-rays [58]. In this review, electromagnetic fields (EMFs) are defined as nonionizing radiation excluding rays of light above 3 THz, such as ultraviolet, visible, and infrared rays. These EMFs are extremely low-frequency (ELF) EMFs below 300 Hz, static electric fields, magnetic fields, and radio waves, including radio frequency (RF) EMFs (10 MHz–300 GHz).

ELF-EMF stimulation has already been demonstrated to induce electrical charges and currents in living organisms [58,59,60]. Therefore, ELF-EMFs can noninvasively apply electrical stimulation and avoid several important problems inherent to electrodes, such as cytotoxicity, tissue incompatibility, and the modification and corrosion of the electrode surface [31]. However, exposure to RF-EMFs results in the generation of heat along with current induction in the target [61,62]. RF-EMFs are currently used in medical diathermy and other applications because they can penetrate the skin surface and generate heat in deep tissue noninvasively [63,64]. Moreover, the nonthermal effects of RF-EMF exposure have also been studied in vitro [65,66,67].

In 1993, Blackman et al. found that ELF-EMFs exert an inhibitory effect on neurite outgrowth at field strengths below 10 µT in PC12 cells stimulated with NGF [68]. Blackman et al. also showed that exposure to a 4 µT, 50 Hz AC EMF for 23 h in the absence of NGF resulted in increased neurite length in PC12D cells, which produce longer neurites in the presence of NGF compared with parental PC12 cells [69]. To our knowledge, this is the first evidence showing the neurite outgrowth-promoting effects of ELF-EMF stimulation. Another study by Ubeda et al. that examined the effects of melatonin on EMF-induced neurite outgrowth also confirmed similar results [70].

Using an original magnetic field exposure system that enables independent control of the vertical and horizontal magnetic fields toward the bottom of the culture dish, Blackman et al. showed that perpendicular magnetic fields in the presence of NGF can enhance neurite outgrowth compared with parallel magnetic fields with NGF in PC12 cells [71].

In relation to the above studies, two independent research teams reported interesting results in addition to showing that static magnetic fields (SMFs) do not affect promoting neurite outgrowth in PC12 cells. At first, Kim et al. showed that when an SMF at 12 mT was applied parallel to the bottom of the culture dish, neurites grew in a direction perpendicular to the direction of the magnetic field compared with the neurites grown in the absence of a magnetic field [72]. Next, Wang et al. also showed that neurite outgrowth of PC12 cells was inhibited in the presence of an SMF (0.23–0.28 T), which was applied perpendicular to the bottom of the culture dish [73]. Therefore, when studies on the control of the direction of neurite outgrowth are performed, it may be necessary to apply a magnetic field parallel to the bottom of the culture dish in normal two-dimensional cultures in vitro because of the properties of the magnetic field.

Several other research groups have exposed PC12 cells to a uniform vertical AC magnetic field relative to the bottom of the culture dish (Table 4). McFarlane et al. showed that exposure to a 50 Hz AC EMF (4.35–8.25 μT) for 23 h during cell differentiation enhanced the NGF-induced neurite length in PC12 cells, whereas a slightly higher EMF (8.25–15.8 μT) did not [74]. A study by Takatsuki et al. showed that an AC 60 Hz EMF (33.3 μT) increased neurite outgrowth in PC12D cells treated with forskolin [75]. Another research group found that an AC 60 Hz EMF (40 μT) [76] and a 50 Hz AC ELF-EMF (1 mT) for 5 days promoted NGF-induced differentiation in PC12 cells [77]. Recently, Nakamachi et al. confirmed that a 50 Hz ELF-EMF (4.2 μT), which was the same stimulation conditions as described by Blackman et al. [69,70,71], promoted the length of neurite outgrowth in PC12 cells using in vitro three-dimensional cell cultures on collagen scaffolds [78]. Thus, these reports suggest that magnetic flux density plays an important role in the biological effects of ELF-EMFs on neurite outgrowth in PC12 cells. They also demonstrate the difficulty in establishing the critical exposure variables (such as magnetic flux density, frequency, or stimulation time) for AC magnetic fields in an extremely low frequency range; however, the exposure variables for pulsed electromagnetic fields (PEMFs) may be more complex, as described below.

Several independent research groups exposed PC12 cells to low-frequency PEMFs parallel to the bottom of the culture dish (Table 4). For example, Fan et al. showed that a 50 Hz PEMF at 0.23 mT significantly increased the number of neurites per cell in PC12 cells treated with NGF [79]. In addition, they also showed that while a 50 Hz PEMF at 1.32 mT decreased the average length of the neurites, the average length parallel to the direction of the PEMF was significantly longer compared with the average length perpendicular to the PEMF. Moreover, Zhang et al. exposed PC12 cells that were stimulated with NGF to a 50 Hz PEMF with different pulse duties at 1.36 mT generated by a pair of 13 cm radius Helmholtz coils [80]. Duty cycles of 10%, 30%, 50%, 80%, and 100% (direct current, hereafter DC) were used in the study. The results showed that the PEMF at a 10% duty cycle significantly reduced the number of neurite-bearing cells but increased the average length of the neurites extending along the direction of the PEMF. However, at a 100% duty cycle, the number of cells with neurites increased and the average length of the neurites decreased. No clear increase in neurite length along the direction of the PEMF was observed at any duty cycle tested, except for the 10% duty cycle. In addition, the same research group confirmed that the application of PEMFs with flux densities at 0.19 and 1.37 mT and a 50 Hz frequency (10% duty cycle) suppressed the percentage of neurite-bearing cells. At the same time, however, these conditions also increased the average length of the neurites compared with the controls [81]. Furthermore, the study also showed that exposure to the PEMF with a flux density at 1.37 mT with a 50 Hz or 70 Hz frequency enhanced the neurite length in the direction of the PEMF.

In addition, Kudo et al. exposed PC12 cells to a PEMF (0.172 Hz, pulse width 4 ms) at a very high flux density (700 mT) perpendicular to the culture dish [82]. One of the interesting findings was that the differentiation of the PC12 cells was induced by the PEMF exposure without adding NGF. Furthermore, the enzyme activity of acetylcholinesterase (AChE), which is a biochemical marker of neuronal differentiation in PC12 cells, was also increased by the PEMF exposure without NGF. In the aspect of molecular mechanism, the authors suggested that this may be attributed to the participation of the MEK–ERK1/2 signaling pathways.

The results described in the above studies clearly suggest that the effect of the employed EMF stimulation on the neuritogenesis of the PC12 cells may largely depend on the experimental conditions of the EMF. One reason why such contradictory data are sometimes observed is that various exposure parameters, such as the intensity, frequency, and type of generated EMF, may affect the cells differently. In fact, in many previous studies, various types of EMF (sinusoidal, pulsed, or a more complex waveform) have been used for such studies [83].

RF-EMF can cause tissue heating (thermal effect), and it sometimes leads to tissue damage or destruction [83]; however, there are a few reports describing the nonthermal effects of RF-EMF. Based on these reports, RF-EMF exposure at intensities that do not cause obvious thermal changes can exert beneficial influences on neuronal differentiation. For example, Inoue et al. examined possible cell differentiation in NGF-hyposensitive PC12 mutant cells (PC12m3 cells) using RF-EMF exposure at 2.45 GHz (200 W) for 60 min at 37 °C as the stimulus [65]. The results suggested that p38 MAPK induces neurite outgrowth through the CREB signaling pathway in PC12m3 cells through RF-EMF stimulation in the presence of NGF. Moreover, Maioli et al. exposed PC12 cells to 2.45 GHz RF-EMF with very weak intensity (400 µW/m^2^) through radioelectric asymmetric conveyer (REAC) technology, which was designed to recover the correct endogenous bioelectrical activity using an asymmetric conveyer probe [66]. As a result, REAC treatment significantly increased neurite length in the PC12 cells. Because of the conveyer (asymmetric probe), the REAC emission could interact with the biological tissues of the human body without depth limits and induce radio frequency microcurrents in tissues that vary based on their molecular characteristics [84]. Thus, REAC has been used for clinical studies of psychiatric and neurological diseases, including Alzheimer’s disease, with encouraging results in the absence of side effects [84,85,86]. Another study also reported increased levels of AChE enzyme activity in PC12 cells exposed to a 1.8 GHz RF-EMF [67].

Taken together, the results of these studies have prompted us to speculate that the effects of various EMF exposure parameters may sometimes act independently on biological processes in PC12 cells; however, more detailed and systematic studies are needed to clarify the underlying mechanism of the effects of EMF on neurite outgrowth.

### 2.5. Optical Stimulation

The application of light in the visible spectrum and near-infrared range is considered a safe and minimally invasive approach that may represent a promising option for treating neurological disorders. Several reports have been published in the context of optical methods for neurite outgrowth in PC12 cells. One study reported that combining NGF with optical approaches may be more effective at promoting neurite outgrowth in PC12 cells (Table 5). For example, Higuchi et al. studied the effect of visible light on NGF-induced neurite outgrowth and observed a higher extension with a short duration (1 min per hour) of intermittent light irradiation (525 nm wavelength) between long episodes of non-irradiation [87]. In another study, the degree of neurite outgrowth under continuous stimulation with different wavelengths of visible light was compared. Enhanced neuronal outgrowth was observed following blue light stimulation with a 470 nm wavelength in NGF-treated PC12 cells [88]. A similar result was observed by Saito et al., who used another wavelength as a stimulus for PC12 cells. Diode laser irradiation with near-infrared light at a wavelength of 810 nm enhanced the activity of p38 MAPK, thus leading to neurite outgrowth in PC12 cells [89]. In addition, another study reported that far-infrared rays promoted neurite outgrowth in PC12 cells in the presence of NGF by activating the AKT1 signaling pathway [90].

A common point of these studies is that small variations in light parameters can alter the effect of light on PC12 cells. In fact, altered light parameters can unexpectedly suppress neuronal differentiation. These results may imply that PC12 cells show a specific response in the neurite outgrowth ratio not only to the wavelength, but also to the intensity and the irradiation time of the light. Interestingly, these studies suggest that the observed effects of optical stimulation are the result of nonthermal effects [87,88,89,90].

Recently, photobiomodulation (PBM) has become a well-known therapeutic method in the medical field for biostimulation. In general, PBM does not use a photosensitizer molecule and works by applying low-level visible or near-infrared light to the target tissue or cells without any adverse effects. The light targets mitochondria and stimulates the production of low ROS concentrations. Depending on its intracellular concentration, ROS triggers various reactions by stimulating signaling pathways. Topaloglu et al. examined the effect of PBM following LED irradiation at a wavelength of 655 nm and found that PBM enhanced NGF-induced PC12 cell differentiation [91]. Thus, light-induced mechanisms with properly optimized light parameters may represent a promising therapy that could induce the differentiation and regeneration of neuronal cells.

### 2.6. Thermal Stimulation

Thermotherapy has been used to treat a variety of diseases for many years, from heating pads to promote blood circulation in peripheral vascular diseases [92] to hot-spring therapy for the treatment of illness and injury [93,94] and even hyperthermia therapy, which may be effective for cancer treatment [92,93,94,95,96,97]. However, there are few in vitro studies examining the effects of temperature change on the process of neuronal differentiation (Table 6).

Kano et al. showed that mutant PC12m3 cells, which do not exhibit neurite outgrowth in response to NGF, extended neurites through activation of the p38 MAPK pathway after heat shock (10 min at 44 °C) alone [98]. Moreover, the study showed that a combination of NGF and heat shock further increased the number of neurite-bearing cells.

Another study reported that PC12 cells that were subjected to a combination of NGF and heat shock (1 h at 42 °C) exhibited increased neurite elongation compared with cells treated with NGF alone [99]. This study also showed that the heat-shock-mediated enhancement of NGF-induced neurite elongation was not accompanied by activation of the p38 MAPK pathway; however, activation of the ERK signaling pathway (via activation of the MEK1/2 pathway) was observed, which was partially mediated by heat shock enhancement of the NGF-induced neurite elongation.

In addition, Kudo et al. first reported that when PC12 cells were treated with temperature-controlled repeated thermal stimulation (TRTS) at 39.5 °C for up to 18 h/day alone, the number of neurite-bearing cells increased through both the ERK1/2 and p38 MAPK signaling pathways [100]. Furthermore, they established two novel subclones: PC12-P1F1, with high sensitivity to TRTS, and PC12-P1D10, with low sensitivity to TRTS [7]. The two subclones were derived from PC12 parental cells based on their TRTS sensitivity during TRTS-mediated neuronal differentiation. Based on the new evidence acquired from the experiments using these novel subclones, the authors suggested that the BMP pathway may be required for TRTS-induced neurite outgrowth [4,7].

Interestingly, Motoda et al. recently reported the differentiation-promoting effects of a contrast bath (thermal stimulation in the combination of warming at 44 °C for 10 min, cooling at 4 °C for 30 min, and warming at 44 °C for 10 min) in NGF-hyposensitive PC12m3 mutant cells [101]. In this context, a contrast bath is a type of rehabilitation therapy, and the affected area is alternately immersed in warm and cold water to improve impaired blood flow and recover from fatigue. The authors also showed that the combination of a contrast bath and NGF markedly promoted neurite outgrowth in PC12m3 cells. In addition, stimulation with a contrast bath alone, without NGF, induced neurite outgrowth in PC12m3 cells, albeit at a low rate. The induction of neurite outgrowth with the combination of NGF and a contrast bath resulted from the activity of p38 MAPK induced by calcium influx into the cells via activated TRP ion channels in the PC12m3 cells, implying that neurite outgrowth in PC12m3 cells induced by the contrast bath alone is also p38 MAPK-dependent.

Taken together, a variety of thermal stimuli cause slight temperature fluctuations in the cells, which may contribute to the effects observed in the reports described above. Unfortunately, precise temperature measurements were not performed in some of the above studies. Thus, a comprehensive understanding of the detailed mechanisms of the various effects of thermal stimulation, including neuronal differentiation, may result in the development of new therapeutic approaches.

## 3. Conclusions

In this comprehensive review, we found that stimulating neuronal differentiation in PC12 cells without relying on diffusible humoral factors such as NGF is already established for use with electrical, electromagnetic, and thermal stimuli but is currently challenging with mechanical, ultrasound, and light stimuli. For the latter three categories of physical stimuli, it will be necessary to develop methods capable of inducing neuronal differentiation with a physical stimulus alone, independent of diffusible factors, to establish a method that could be applied to the human body in the future. With respect to the physical stimuli addressed in this review, much remains unknown regarding the mechanisms underlying neuronal differentiation induction, including the receptors activated and the associated downstream intracellular signaling pathways. In the future, elucidating the entire mechanism for each physical stimulus will enable the application of these respective methods to induce neural regeneration in the human body. This will contribute to the development of more efficient neuroregenerative medical technologies using neural stem cells.

## Figures and Tables

**Figure 1 ijms-25-00772-f001:**
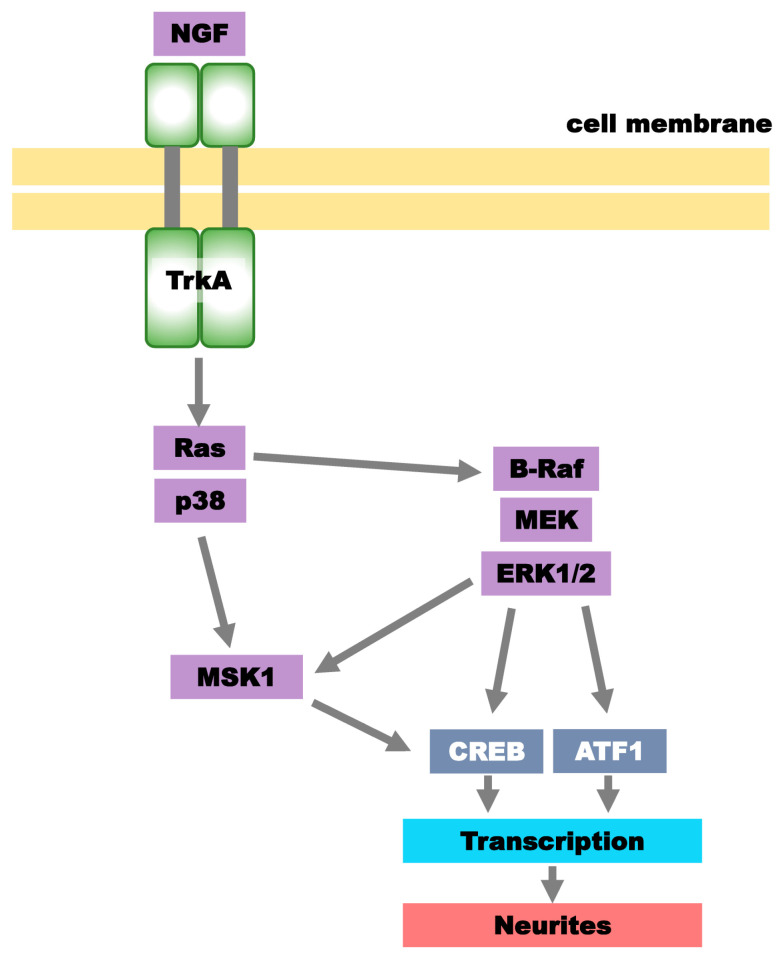
A schema for the principal signaling pathways involved in NGF-induced neuritogenesis in PC12 cells. NGF binds to and activates TrkA, a receptor for NGF. The activated TrkA then stimulates Ras, which can activate both the ERK1/2 and p38 MAPK signaling pathways. These two pathways are essential for the NGF-induced neuronal differentiation of PC12 cells. MSK1 is a target molecule of NGF-mediated ERK1/2 and p38 MAPK. CREB and ATF1 may serve as representative transcription factors mediating the transcription processes for neuritogenesis induced by NGF. NGF, nerve growth factor; TrkA, tropomyosin-related kinase A; Ras, rat sarcoma virus; B-Raf, v-raf murine sarcoma viral oncogene homolog B1; MEK, mitogen-activated protein kinase; ERK1/2, extracellular signal-regulated protein kinases 1/2, MSK1, mitogen- and stress-activated protein kinase 1; CREB, cAMP response element binding protein; ATF1, activating transcription factor 1.

**Table 4 ijms-25-00772-t004:** The enhancing effect of electromagnetic field stimulation on neuronal differentiation in PC12 cells and their derivatives in the presence or absence of an additional condition.

Electromagnetic Stimulation Conditions				Cell Line	Additional Condition for Inducing Differentiation	Detected Changes during Differentiation	Reference
Stimulation Pattern	Frequency	Intensity	Stimulation Term				
Sinusoidal ELF	50 Hz	2.2–4 µT	22 h	PC12D	Not used	a	[69]
	50 Hz	4 µT	23 h	PC12D	Not used	a	[70]
	45 Hz	20.2 µT	23 h	PC12	NGF	a	[71]
	50 Hz	4.35–8.25 μT	23 h	PC12	NGF	a	[74]
	60 Hz	33.3 μT	22 h	PC12D	Forskolin	a	[75]
	60 Hz	40 μT	22 h	PC12D	Forskolin	a	[76]
	50 Hz	1 mT	5 days	PC12	NGF	a, b, c	[77]
	50 Hz	4.2 μT	7 days	PC12	NGF	a	[78]
Low-frequency PEMF	50 Hz	0.23 mT	96 h	PC12	NGF	a	[79]
	50 Hz (10% duty cycle)	1.36 mT	96 h	PC12	NGF	a	[80]
	DC	1.36 mT	96 h	PC12	NGF		
	50 Hz (10% duty cycle)	0.19 mT	96 h	PC12	NGF	a	[81]
	50 Hz (10% duty cycle)	1.37 mT	96 h	PC12	NGF		
	70 Hz (10% duty cycle)	1.37 mT	96 h	PC12	NGF		
	0.172 Hz (pulse width: 4ms)	700 mT	12 h/day	PC12	Not used	a, c	[82]
Radio frequency	2.45 GHz	200 W	60 min	PC12m3	NGF	a, c	[65]
	2.45 GHz	400 µW/m^2^	96 h	PC12	Not used	a, b, c	[66]
	1.8 GHz	40 W	24 h	PC12	Not used	b, c	[67]

ELF: extremely low-frequency; NGF: never growth factor; PEMF: pulsed electromagnetic field; DC: direct current. a: morphological changes with neuritogenesis. b: changes in gene expression. c: changes in protein expression and/or activity.

**Table 5 ijms-25-00772-t005:** The enhancing effect of optical stimulation on NGF-induced neuronal differentiation in PC12 cells and their derivatives.

Optical Stimulation	Wavelength	Optical Stimulation Conditions	Additional Condition for Inducing Differentiation	Detected Changes during Differentiation	Reference
Visible light	525 nm	0.4 mW/cm^2^, 1 min/h	NGF	a	[87]
Visible light	470 nm	1.8 mW/cm^2^, 36 h	NGF	a	[88]
Near-infrared light	810 nm	5–20 J/cm^2^ (10 W, 1.26–5.04 S)	NGF	a, c	[89]
Far-infrared rays	5–12 μm	3.1 mW/cm ^2^, 30 min/day	NGF	a, c	[90]
Visible light	655 nm	1 J/cm^2^ (50 mW, 251 s)	NGF	a, b	[91]

NGF: never growth factor. a: morphological changes with neuritogenesis. b: changes in gene expression. c: changes in protein expression and/or activity.

**Table 6 ijms-25-00772-t006:** The enhancing effect of thermal stimulation on neuronal differentiation in PC12 cells and their derivatives in the presence or absence of an additional condition.

Thermal Stimulation Conditions	Cell Line	Additional Condition for Inducing Differentiation	Detected Changes during Differentiation	Reference
44 °C, 10 min	PC12m3	Not used	a, c	[98]
44 °C, 10 min	PC12m3	NGF		
42 °C, 1 h	PC12	NGF	a, b, c	[99]
38.7 °C, total 18 h/day	PC12	Not used	a, c	[100]
Near 39 °C, total 18 h/day	PC12-P1F1	Not used	a	[7]
Near 39 °C, total 18 h/day	PC12-P1F1	SP600125	a, b	[4]
Near 39 °C, total 18 h/day	PC12-P1F1	BMP4		
10 min for 44 °C/30 min for 4 °C	PC12m3	Not used	a, c	[101]

NGF: never growth factor; BMP: bone morphogenetic protein. a: morphological changes with neuritogenesis. b: changes in gene expression. c: changes in protein expression and/or activity.

## Data Availability

The data presented in this study are available in the article.

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
