# Peer review of "Physical Stimulation Methods Developed for In Vitro Neuronal Differentiation Studies of PC12 Cells: A Comprehensive Review"

_ijms, 2024, doi:10.3390/ijms25020772_

Round 1

Reviewer 1 Report

Comments and Suggestions for Authors

     This manuscript presents a detailed summary of present knowledge in the field of PC12 cell neuritogenesis induced/affected by various physical stimuli (mechanical, electrical, electromagnetic, optical, thermal stimuli). Whereever data are available, the possible involvement of intracellular signaling mechanisms is also discussed. The authors evaluate the effects of various treatment protocols and conclude that small variations of stimulation parametersmay have dramatic effects on the outcome of stimulation. They also state that some of the physical treatments induce neurite formation by themselves, while others influence neurotrophin-stimulated differentiation only. The review is comprehensive (as stated in the title), well-organized (the tables are especially useful). It mostly targets experts in the field, may be too much for the general audience.

      A few errors that need to be corrected:

lines 26 and 68: "...p38MAPK signaling mediated by ERK1/2..."  This is not true: the p38MAPK and ERK1/2 pathways are parallel pathways that may cross-talk. The p38MAPK pathway is mediated by p38MAPK (as shown in Figure 1).

line 56:  MSK1 is a protein kinase, not a transcription factor.

line 462:  The abbreviation p38MAPK was used several times before, it is not necessary to explain it again.

line 512:  Remove underlining of signaling pathways.

Reviewer 2 Report

Comments and Suggestions for Authors

The paper is devoted to the review of studies on stimulation of neuronal differentiation of the PC12 line using various biophysical effects. The authors provide data on the effect of mechanical stimulation, ultrasound, electrical and electromagnetic stimulation, optical and thermal effects. The review seems to be quite extensive and exhaustive. Undoubtedly, it is of interest to specialists who use this model in their experiments.

It would be useful for this publication to specify the phrase "neuronal differentiation" in the review tables. Perhaps it would be convenient to introduce an additional column in the tables that would indicate the specific action of a certain factor - whether it was only a change in cell morphology, or whether activation of kinases, signaling cascades, and other events accompanying differentiation was also detected. Perhaps, this will allow us to see the specificity of the action of each of the mentioned effects, as well as to make hypotheses about the mechanism of their action.

This work can be recommended for publication.
